# Accessing medicines for non-communicable diseases: Patients and health care workers' experiences at public and private health facilities in Uganda

**Andrew K. Tusubira** [1]\*, **Ann R. Akiteng**[1], **Brenda D. Nakirya**[1], **Ritah Nalwoga**[1], **Isaac Ssinabulya**[1,2,3], **Christine K. Nalwadda**[1,4☯‡], **Jeremy I. Schwartz** [1,5☯‡]

1 Uganda Initiative for Integrated Management of Non-Communicable Diseases, Kampala, Uganda, 2 Uganda Heart Institute, Mulago National Referral Hospital, Kampala, Uganda, 3 Department of Medicine, Makerere University College of Health Sciences, Kampala, Uganda, 4 Department of Community Health and Behavioural Sciences, School of Public Health, College of Health Sciences Makerere University, Kampala, Uganda, 5 Section of General Internal Medicine, Yale University School of Medicine, New Haven, CT, United States of America

☯ These authors contributed equally to this work.
‡ These authors are joint senior authors on this work.
\* andrewtusu@gmail.com

**Data Availability Statement:** All relevant data are within the manuscript and its Supporting Information files.

## Abstract

### Background

Non-communicable diseases (NCDs) are increasingly prevalent in low- and middle-income countries. Successful management requires consistent access to appropriate medicines. Availability of NCD medicines is generally low, especially in the public sector, however, little is known about other factors affecting access. We explored barriers and facilitators of access to medicines for diabetes and hypertension at public and private health facilities in Uganda.

### Methods

We conducted a qualitative descriptive study at six public hospitals and five private health facilities in different regions of Uganda. Data collection included 36 in-depth interviews and 14 focus group discussions (n = 128) among purposively selected adult outpatients with diabetes and/or hypertension and 26 key informant interviews with healthcare workers and patient association leaders. Transcripts were coded and emerging themes identified using the Framework method.

### Results

Four main themes emerged: Stocking of medicines and supplies, Financial factors, Individual behaviour and attitudes, and Service delivery at health facilities. Stocking of medicines and supplies mainly presented barriers to access at public facilities including frequent stock-outs, failure to stock certain medicines and low quality brands often rejected by patients. Financial factors, especially high cost of medicines and limited insurance coverage, were

**Funding:** This work was supported by the Doris Duke Charitable Foundation International Clinical Research Fellowship Pilot Twinning Award granted to AKT. The funders did not play any role in the study design, data collection and analysis, decision to publish and in the preparation of this manuscript.

**Competing interests:** The authors have declared that no competing interests exist.

barriers in private facilities. Free service provision was a facilitator at public facilities. Patients' confusion resulting from mixed messages and their preference for herbal treatments were cross-sector barriers. While flexibility in NCD service provision was a facilitator at private facilities, provider burnout and limited operating hours were barriers in public facilities. Patient-driven associations exist at some public facilities and help mitigate inadequate medicine stock.

## Conclusion

Access to NCD medicines in Uganda is influenced by both health system and patient factors. Some factors are sector-specific, while others cross-cutting between public and private sectors. Due to commonalities in barriers, potential strategies for overcoming them may include patient-driven associations, public-private partnerships, and multi-modal health education platforms.

## Background

Non-communicable diseases (NCDs) now account for the majority of the global burden of disease, which disproportionately impacts low- and middle-income countries (LMIC) [1, 2]. The World Health Organization (WHO) estimates that by 2020, 80% of global deaths will be due to NCDs. In sub-Saharan Africa (SSA) alone, the NCD burden is expected to increase by 27% in that time frame [3].

Medicines represent a critical component in the management of NCDs. The WHO Global NCD Action Plan calls for 80% availability of medicines used to treat NCDs by 2020 [3]. However, availability in LMIC remains well below this target. Recent studies in Uganda have reported low availability, and a high degree of disparity in the availability, of most medicines essential to the management of common conditions such as diabetes and cardiovascular diseases [4, 5].

Jacobs, et al characterize four domains of access to health services, namely availability, affordability, geographical access, and acceptability [6]. Aside from the documented poor availability, sparse data exist about other barriers faced by patients when attempting to access NCD medicines. It is important to understand the current challenges and the context in which NDC patients are accessing medicines in order to provide suggestions for policy and programmatic options. Additionally, information that offers appropriate responses to the demands for NCD medicines will promote equitable access to NCD medicines at both public and private health facilities in the country. We therefore conducted this qualitative study to explore both barriers and facilitators of access to medicines to treat diabetes and hypertension at public and private health facilities in Uganda.

## Methods

### Study setting

We conducted this study at public hospitals and privately owned health facilities in Uganda. Public facilities in Uganda are government owned and provide free healthcare services, including the dispensing of medicines from facility-based pharmacies. The public sector health care system works on a referral basis, with health facilities categorized in levels including Health Centre Level I, II, III, and IV, followed by General Hospitals (GH), Regional Referral Hospitals

(RRH), and the National Referral Hospital (NRH). All hospitals (public or private) are expected to offer NCD care [7]. The private health sector in Uganda is varied and diverse. The private-not-for-profit (PNFP) sector mainly includes faith-based organizations. PNFPs offer healthcare services at relatively lower prices compared to other private healthcare providers partially because of the financial subsidies provided to PNFPs by the government [8]. The private-for-profit (PFP) sector includes clinics, pharmacies, medical centres, and hospitals [9]. Services provided at PFP facilities are financed through direct payment or private insurance schemes. Of the 6,937, health facilities and clinics in Uganda, 3,133 (45.16%) are government owned (public); 2,976 (40.31%) PFP; and 1,008 (14.53%) PNFPs. The public sector and PNFPs are mostly higher level health facilities including the two NRHs and all the 14 RRHs that are public. The PFPs majorly consist of lower level HC IIs and clinics [10]. Although most facilities do not track the number of NCD patients, to provide a clear distribution NCDs care at public or private facilities, NCD service provision in the Uganda is highly concentrated at the public hospitals and most NCD patients attend public hospitals, bypassing nearby lower-level facilities [11, 12]. Besides, lower-level facilities also routinely refer patients with hypertension and diabetes to public hospitals [7]. There is no public-sector health insurance scheme currently in Uganda.

## Study design

This was a qualitative descriptive study conducted between December 2016 and May 2017 using in-depth interviews (IDI), key informant interviews (KII), and focus groups discussions (FGD). We used IDI to elicit individual lived experiences of patients in accessing medicines for hypertension and diabetes and FGD to explore shared experiences, norms, and health-seeking behaviour among these patients. We conducted KII with prescribers and dispensers to elicit their perceptions on barriers and facilitators to accessing medicines at the health facilities.

## Study participants

We initially recruited patients and healthcare workers (HCW) to participate in this study. Patient participants were adults with diabetes, hypertension, or both conditions attending outpatient clinics in the sampled health facilities. Patients with a mental disability were excluded. HCW were eligible to participate if they had responsibility for prescribing and/or dispensing medicines to treat diabetes or hypertension at the respective health facility. During data collection, we learned about the existence of informal but organised diabetes patient groups at some public hospitals. These groups had a chairperson, who provided leadership responsibilities including mobilizing funds from group members and buying for them medicines. We t therefore included chairpersons of these groups among our participants because they would provide us with deeper information about the operation of these groups, which other patients would not provide.

## Sampling strategy

We purposively selected six public hospitals, two PNFP hospitals, and three PFP medical centers (Table 1). The selected health facilities are located in different regions and districts of Uganda. We selected the public facilities following geographical diversity. We selected four RRHs of the 14 in the country. These were: Gulu RRH (northern Uganda); Mbarara RRH (western Uganda); Fort Portal RRH (western Uganda—Rwenzori region); and Jinja RRH (eastern Uganda). We then selected two GHs in the central region: Entebbe GH (Wakiso district) and Naguru GH (Kampala). We selected two PNFP hospitals located in two different

**Table 1. Selected health facilities, interviews and discussions conducted.**

| Public Hospitals | IDI (N = 36) | FGD (N = 128) | KII (N = 26) |
|---|---|---|---|
| Fort Portal RRH (Buhinga) | 5 | 2 FDG (n = 16) | 3 |
| Naguru GH | 4 | 2 FDG (n = 20) | 2 |
| Entebbe GH | 5 | 2 FDG (n = 16) | 3 |
| Mbarara RRH | 4 | 2 FDG (n = 24) | 3 |
| Jinja RRH | 6 | 2 FDG (n = 16) | 3 |
| Gulu RRH | 2 | 2 FDG (n = 20) | 2 |
| **Private Health Facilities** | | | |
| Lubaga Hospital | 4 | 2 FDG (n = 16) | 3 |
| Kabarole Hospital | 2 | 0 | 3 |
| IMCS | 4 | 0 | 4 |

RRH = Regional Referral Hospital; GH = General Hospital; IMC = International Medical Centre.

regions, which included Kabarole hospital (Western Uganda) and Lubaga Hospital (Central—Kampala). The three PFP facilities were all within the International Medical Centre (IMC) network and included Park Royale, Kitgum House, and Kololo, all in Kampala (Since IMC service procedures are the same across all the centers in the country). IMC is one of the largest PFP networks in the country with 13 centers. We selected more public facilities than private because provision of NCD care is highly concentrated at public hospitals[12].

We used maximum variation sampling strategy in the selection of our study participants. We purposively selected patients living with diabetes and/or hypertension receiving care at public facility NCD clinics and private facility general outpatient clinics as well as HCW providing care at these facilities. Patients were identified and informed about the study prior to receiving care. Following their clinician encounter, selected patients were invited to either participate in an FGD or an IDI. At least two key informants were selected from each health facility. We also purposively selected two chairpersons of diabetes patient groups, one from each of the two facilities where these groups operate, due to their leadership roles and long-time experience in mobilising patients.

## Data collection and management

In order to obtain a wider range of information from each site, we pre-set to conduct at least four IDI, two KII and two FGD at each selected facility. We therefore conducted 36 IDI, 14 mixed-gender and not disease specific FGD (8–12 participants per group; total n = 128) and 26 KII. All interviews and FGD were conducted at the sampled facilities. We were not able to conduct FGDs at some sites because of the very low turn-up of NCD patients per day, which could not support mobilization of participants for FGD. Prior to data collection, we pretested the interview guides among four patients with diabetes and hypertension and three HCW. The pre-tests were conducted in a manner that replicated the anticipated flow of data collection sessions, including administration of consent forms, demographic questionnaire, and interview guide. We then revised the study materials to ensure that questions were clear, appropriate, and aligned with the overarching research question.

The pre-tested IDI and FGD guides included the following key topics: patients' account of medical history, healthcare-seeking behaviour, and impression of service provision at that facility. We asked about experiences and challenges faced in obtaining prescribed medicines and patterns of resort when facing challenges with this. KII guides included questions about HCWs' experience in NCD work, NCD service provision at that facility, their perspectives on

patients' uptake of NCD services, and challenges they face when providing NCD services (see S1 File). IDI and FGD guides were translated into three local languages commonly spoken among persons living at the different study sites: Luganda, Runyankole, and Acholi. Three trained and experienced research assistants conducted interviews and moderated discussions under the supervision of AKT. At each facility, we first conducted IDI and later held the FGD. A research assistant would invite a patient to participate in what was being conducted at that time, either to be interviewed or to be part of an FGD. A patient would participate in only one of these. IDI and FGD were conducted in local languages most commonly spoken in the respective areas, audio recorded, and observations were recorded throughout. Interview or discussion time ranged between 30 and 70 minutes. Audio recordings were transcribed verbatim. IDI and FGD recordings were translated into English after verbatim transcription. All transcripts were proofread and compared against their recordings to check for accuracy of translation, completeness of transcription, and to ensure data quality before analysis.

## Data analysis

We analysed the data using a thematic approach following the Framework Method for the analysis of qualitative data [13]. The Framework Method is appropriate for thematic analysis where it is vital to compare and contrast data by themes while still placing each perspective in its context. In this strategy, a series of interconnected stages are followed that enable a research team to move back and forth across the data, while making cross-linkage between codes and categories, and linking the original data with the findings. Themes are therefore generated by making comparisons within and between cases, while retaining the original meanings and feel of the participants' words. Usually, each category falls under a given sub-theme or theme [13].

Eight transcripts were first coded by a multidisciplinary team that included members with expertise in public health, social sciences, and medicine (AKT, BND, RN, ARA). Each member independently developed initial codes which were discussed and refined into a working analytic framework during in-person meetings. Using Atlas.ti (version 7.5.7), AKT applied this framework to all transcripts while allowing new codes to emerge, leading to a final framework that was agreed upon by the team. We used this final framework to develop categories and, subsequently, themes. Our methods conform to the standardized COREQ approach (S2 File).

## Ethics and consent

The study was approved by Makerere University School of Medicine Ethics Review Committee, Uganda National Council of Science and Technology (number: SS 4415), and Yale Human Subjects Committee. Administrative clearance was obtained from all sampled health facilities. Written informed consent, including study purpose and permission for audio recording, was obtained from participants. Illiterate participants provided consent through thumbprints after being explained to and read the consent form to them.

## Results

### Participant characteristics

We enrolled 164 patient participants, of whom 138 (84%) were at public facilities. Their mean age was 55 years (standard deviation ±9.7 years) and 106 (65%) were female (Table 2). We enrolled 24 HCW including 12 prescribers (10 medical doctors and two clinical officers) and 12 dispensers (eight pharmacists and four clinical officers). Two diabetes patient association chairpersons were also enrolled.

**Table 2. Patient participant characteristics.**

| Characteristic | IDIs (n = 36); n (%) | FGDs (n = 128); n (%) |
|---|---|---|
| **Age** | | |
| 30–44 | 5 (22.7) | 17 (77.3) |
| 45–54 | 11 (24.4) | 34 (75.6) |
| 55–64 | 15 (23.4) | 49 (76.6) |
| 65–75 | 5 (15.2) | 28 (84.8) |
| **Gender** | | |
| Female | 24 (22.6) | 82 (77.4) |
| Male | 12 (20.7) | 46 (79.3) |
| **Health facilities** | | |
| Public facilities | 26 (18.8) | 112 (81.2) |
| Private facilities | 10 (38.5) | 16 (61.5) |

Four main themes describing access to medicines emerged in our analysis: Stocking of medicines and supplies; Financial factors; Individual behaviours and attitudes; and Service delivery at health facilities (Table 3).

Herein, we present representative quotations for each theme, noting interview type, facility sector, and location following each quotation. For patient participants, we also note gender and age.

## Stocking of medicines and supplies

Stocking of medicines and supplies was largely a barrier to accessing medicines. Frequent stockouts represented the commonest barrier at public facilities. Patients at public facilities often described receiving partial doses or none of the prescribed medicines. In both FGD and IDI, these patients complained that even when they returned to the health facility for medicine refills, medicines were often out of stock. An excerpt from a public facility FGD (Jinja) illustrates this barrier:

*There are times when we spend about two to three months without receiving medicines. Like by the end of last month, we had spent two months without receiving medicines.* (Public, female, 46 years).

*For some on insulin, there can be like 10 syringes in a pack but without any bottle [of insulin].* (Public, female, 50 years).

*What brings more problems is when the test strips are out of stock, yet it is a must we have to take the blood test before we get medicines and the doctor will not work on you before your blood is checked.* (Public, male, 61 years).

Another patient stated:

*I have spent almost one year without getting a complete dose of medicines. Whenever I come, they tell me to go and buy.* (IDI, Public, male, Gulu, 43 years).

Failure to stock certain essential NCD medicines also represented a barrier at both public and private facilities. This was mainly expressed by HCWs at both public and private facilities who reported that they often had few medicines from which to choose when prescribing, since their facility pharmacies fail to stock a variety of options.

**Table 3. Summary of the coding framework.**

| Example of Codes | Category | Theme |
|---|---|---|
| Partial dose dispensed, Limited drug option, only basic medicines, run out of priority medicines, prescribe what is available, patients aware of drug shortage, no refills, discharge without medicine | Frequent stock outs *Both (Barrier) | Stocking of medicines and supplies |
| No Medicines for advanced disease, few medicines for complications, medicines not easily found on market, medicines not procured, not common drugs | Medicines not stocked *Provider (Barrier) | |
| Complaints on quality of medicine, complaints on side effects, Refuse medicines, affect appetite, | Perceived quality of stocked brands *Both (Barrier) | |
| Some medicines dispensed only to: vulnerable patients, the elderly, dispensed to a few | Rationing medicines*Provider (Facilitator) | |
| Often told to buy, unable to buy, long process getting money, transport challenges, look for cheaper pharmacies, buying drugs in bits, no funding, half dosage payment, long distance travelled | Financial constraints / no money *Both (Barrier) | Financial factors |
| Expensive medicines, Low insurance allocation, expensive tests, cash patients few, unaware of medicine cost, medicines expensive, transfer to public facilities | Payment for private services *Both (Barrier) | |
| Free medicines, no consultation fee, insurance increased access | Free medication and insurance facilitators access *Both (Facilitator) | |
| Hospital medicines ineffective, does not swallow particular medicines, purchased drugs more effective, hospital medicines unsafe, free medicines do not prolong life | Patients' perceptions *Both (Barrier) | Individual behaviours and attitudes |
| Consult different physicians, influence from friends/peers, move from one facility to another, seeks alternative care | Multiple consultation *Both (Barrier) | |
| Mix herbs and hospital meds, prefer herbal, herbs are stronger, less side effects, herbs easily accessed, bitter vegetables a substitute | Herbs an alternative medication *Patient (Barrier) | |
| Specialized clinics, high volume, long waiting, no health education, partial tests, rely on government supply, Pharmacy Serves both out & inpatients, limited time, No follow up, limited NCD care | Management and operation aspects *Patient (Barrier) | Service delivery at health facilities |
| Long waiting, divided prescriber attention, health worker absenteeism, doctors not easily accessed, Interns Vs Staff commitment, negative attitude of HCWs, staff come late, prescription unknown, tests results not explained | Patient dissatisfaction *Patient (Barrier) | |
| Staff exhaustion, burnout, few competent staff, no staff motivation, late presentation issues, Insufficient time for patients, limited work space, less care offered | Staff challenges *Provider (Barrier) | |
| Enough patients' time, access care anytime, unfixed clinic, health workers on time, team work, staff always available | Flexibility in service provision *Both (Facilitator) | |
| Subsidize medicines for members, register members, offer counselling, work with hospital management, health education, advocacy, seek to influence patent care | Role of patient groups *Both (Facilitator) | |

(Symbols used to show main source of content / perception)

*Patient = Patient experience, *Provider = Health provider perception *Both = Both patient and provider perception.

*At times, I prescribe a drug but when it is not part of our stock, so I change prescription to fit in what the patient can get.* (KII, Prescriber, Private, Kabarole).

*But even now our hands are tied! Most of these medicines* [for complications] *are never stocked. It is also difficult to obtain some of them from the open market. That is why we try to restrict ourselves by prescribing what is available unless the condition is worse.* (KII, Prescriber, Public, Mbarara).

Another barrier we identified under this theme was the stocking of particular brands of medicines perceived to be of low quality. This was largely in reference to insulin. Patients with diabetes at public facilities complained of low quality insulin brands being supplied.

*I already told you, that type of insulin is ineffective and with severe side effects. It always affected me and now I just do not want to use it. It does not work for us, not me alone!* (IDI, Public, Gulu, female, 58 years).

*. . .even when we go to other districts, that is the same brand of insulin we will find. We are tired of it but only use it as a last resort when we have no alternative.* (FGD, Public, Kampala, female, Age 60*).*

HCWs were concerned that this perception led to low acceptability, misuse, and rejection of this particular insulin brand by some patients, despite a lack of available alternatives:

*The insulin we have. . . patients are complaining a lot that it is weak and they tend to give themselves larger doses for it to work. So those who can afford, we tell them to look for other brands which are more effective.* (KII, Pharmacist, Public, Kabarole).

The only facilitator identified within this theme was the rationing of medicines for certain patients. At one public facility, HCW reported reserving some basic medicines for a few patients they deemed vulnerable. This approach facilitated access to medicines for a few, typically elderly, patients as narrated by one provider:

*Actually it has been so challenging with certain patients especially the very old patients. . .With the help of the pharmacist, there are some stocks which are reserved for them. Even when we run short of medicines, there are a few which are left for such patients. When these patients come, we give them some of the reserved medicines to keep them going as we wait for the next supply. (*KII, Prescriber, Public, Gulu).

## Financial factors

We found that financial factors, including cost of both medicines and transportation, were largely a barrier to accessing medicines at both public and private facilities. The provision of free medicines at public facilities was, however, a facilitator. Health insurance at PFP facilities was identified as both a barrier and facilitator.

At public facilities, patients were often told to buy prescribed medicines that were out of stock or unstocked, but some reported being unable to do so. Some patients said they would buy medicines in piecemeal when they had money, while others who could not afford them would remain without medicines:

*I walk back home and explain to my people that I have not got medicines. If there is no money, I will fail to buy the drugs and wait to come back* (IDI, Public, Kampala, female, 61 years).

A HCW also noted:

*Some patients come back after a month when they did not buy the medicines we prescribed. Genuinely, some cannot afford these medicines mainly because they do not have money. I have seen them. . .you find a patient has not been getting any medicine from the previous four visits, he does not have money and the patient tells you, 'fine let me go home and I die'.* (KII, Prescriber, Public, Kampala).

High cost was a barrier to accessing medicines at private facilities as well, although some patients opted to seek care at these facilities due to perceived higher quality services. During one FGD at a PNFP facility, patients elaborated:

*Money is the main challenge. . .Whenever we come, we first pay money then we are given a form to proceed to the tests. After the tests, you see a doctor who prescribes and also gives a bill*

*for the medicines. So we find it expensive to maintain such costs, yet we have to take medicines without breaks.* (FGD, Kampala, female, 55 years).

*Hypertension medicines are so expensive. A lot of money is needed which we sometimes do not have. At times they prescribe for you medicines worth [approximately 30 US Dollars]. So we often fail to get the full dose.* (FGD, Kampala, male, 48 years).

A HCW at the same facility further supports this notion:

*Indeed, some drugs are expensive yet most of our patients are poor. We* [HCWs] *also fail to know what most patients do when they do not have money. Our prices are fairer compared to outside pharmacies, but still the medicines are expensive for our patients.* (KII, Prescriber, Kampala).

The provision of free medicines at public facilities motivated patients to seek care there, as illustrated by one patient:

*We mainly come here. . .because we receive free health care. The doctors will not ask us for money. . . paying for services at the privates is a big challenge for most of us.* (FGD, Kabarole, female, 56 years)

However, the cost of transport was cited as a barrier by some who had to travel far distances to obtain care at public facilities, resulting in missed appointments and refills.

*I did not have medicines because I had spent two months without coming for treatment. I could not afford the transport to bring me here.* (FGD, public, Mbarara, male, 68 years).

Health insurance facilitated access to medicines at PFP facilities, among those who were insured:

*To a great extent, insurance has increased access to quality NCD medicines especially for those in urban settings who can access it. Because if they were to pay for treatment each time they visited, it would be a challenge to adhere to treatment plans which would compromise disease management.* (KII, Pharmacist, Kampala).

However, those with limited or expired insurance coverage would face high costs once their coverage ceiling was reached, as narrated by a HCW and a patient:

*Insurance companies have a limit on how much a patient's treatment should cost per visit. This includes all services that the patient receives that day, such as laboratory tests, consultation, medicines, etc. In case the bill exceeds the patient's limit, some items must be cut to reduce the cost and most of the time it is the drugs. . .Also, [medicines and other clinical fees are not covered] if the insurance period expires or money on the insurance card runs out. Even if they have been on this insurance for long.* (KII, Pharmacist, Kampala).

*The challenge with us on insurance is when your insurance does not cover certain medicines, we have to look for them from other pharmacies which is at a cost.* (IDI, female 54 years).

## Individual behaviours and attitudes

Individual behaviours and attitudes were identified only as barriers to access and included patients' confusion resulting from mixed messages, negative perceptions towards free medicines, and a preference for herbal treatments over conventional medicines.

Several patients were noted to be engaged in making multiple consultations for their conditions, including moving from one health facility to another, getting conflicting advice from different providers, and receiving misinformation from friends. These mixed messages resulted in confusion on the part of patients, thereby affecting adherence to treatment.

*I was diagnosed with diabetes at this hospital but I also visit two other private facilities where they monitor my sugar level and tell me about diet. But sometimes I get different advice from these facilities and some medicines are not the same*! (IDI, Public, Mbarara, male, 56).

*I talked to my friend who told me that the medicine I am taking is very dangerous. She told me that I was not supposed to take that type of medicine. . .unless I want to become blind. So I no longer swallow that drug.* (FGD, Public, Kampala, female, 61 years).

Two HCW supported this notion, stating:

*Yes, some patients consult other physicians which may not be wrong, but they come back with mixed ideas about their treatment.* (KII, prescriber, PFP, Kampala).

*At times a patient returns with a very high sugar level then he will confess to me that, 'honestly, I was off drugs for a month because they told me there is some other drug that cures diabetes. I tried it out, but the drug do not work for me'.* (KII, Prescriber, Public, Kampala).

Another attitude exhibited by patients that resulted in a barrier to access was the perception that freely dispensed medicines are less effective than those sold in private pharmacies:

*I think that these free medicines are just to calm the disease but they are not that strong to sustain my life for long. So I swallow them but even if I do not swallow, there will not be a bad effect!".* (FGD, Public, female, 46 years).

Furthermore, patients at public facilities frequently mentioned using both herbal treatments and conventional medicines to manage their conditions. Patients used herbs for numerous reasons, including their ready availability and as an alternative for missed prescriptions or for medicines with strong side effects. Also, some believed that bitter vegetables reduce sugar levels and that certain types of vegetables are more effective than the conventional medicines.

*For my sugar to lower like that, I use herbal. . . the drug that they prescribe for me used to give me terrible side effects. I even came back and complained but [the physician] told me to continue taking that drug and yet it was going to kill me. So I abandoned it and. . .bought herbs. I got the same medicine today, but to tell you the truth, am not going to take it! Because it is the same drug which almost plucked out my eyes.* (FGD, Public, Kampala, female, 60 years).

*When I fail to get medicines, I look around for herbs because they also reduce the sugar. If the sugar levels are high, you can squeeze 'omululuuza', and drink that juice. When your eyes have got a fog like thing, you squeeze 'oluwawu' and drink it.* (IDI, Public, Jinja, female, 50 years).

## Service delivery at health facilities

We found that there were various features of service delivery that affect access to medicines. In public facilities, the barrier was the limited operating hours of NCD clinics, while more

generous operating hours in the private sector facilitated access to medicines. Innovative patient-driven diabetes groups were identified as a facilitator of access in public facilities.

Management of diabetes and hypertension at public facilities is provided within specialised clinics that are limited to a specific day of the week, resulting in high patient volumes, long waiting times, and limited consulting time with providers.

*One day I felt very sick and I came to this clinic only to find HIV patients. The patients were so many and the HCW told me that madam, sorry, on these days we cannot work on you.* (FGD1, Public H/F, Jinja, female, 46 years).

*We come very early even before 6:00am but they start working when it is late. We are kept here in long queues, hungry, and weak. Imagine, I was among the first ten patients to arrive by 6:00am but I interfaced with the doctor at 1:00pm. What time will the last patients see the doctor? Do you think they will see him?* (FGD, Public, Jinja, female, 46 years).

These features also lead to burn-out and absenteeism among providers which hinder patient access to medicines, as evidenced by these participants:

*The patients are many, yet few staff. We are overwhelmed and often get exhausted. By the time I have seen fifty patients, I am already losing attention and the rest of the patients will not get the best from me.* (KII, Prescriber, Public, Kampala).

*When that clinic day comes, we (HCW) come when we are ready to burn out! We actually do not give quality care to these patients because they are too many. I do not have enough time to listen to all medical complaints, explain prescriptions, and add some health education for each patient.* (KII, Prescriber, Public, Kabarole).

Contrary to what was reported at public facilities, we found that NCD care at private facilities was provided every day and patients would access medical care whenever they wanted. Patients reported that such flexibility in service provision enabled access to treatment.

*We are attracted by the customer care at this facility and the attention they give us. I feel the basawo [HCW] here are more caring, friendly and my condition is well understood when I explain to them.* (IDI, PFP, Kampala, female, 57 years).

*Patients come here not because we are the best but we are the more organized and we always serve them better. When you pay, the best selective services are given to you.* (KII, PNFP, Prescriber, Kampala).

In some public facilities, patient-driven diabetes associations have formed and serve as a facilitator to higher quality diabetes care. These associations are informal, yet organised, have a leadership structure, and register their members. Association leaders reported that these groups predominantly sought to handle issues that affect the management of patients with diabetes at the respective facilities, purchase diabetes medicines and supplies, and sell them to members at subsidized prices. Association leaders offer health education, social support, and advocate for their members' needs.

*This association has helped a lot. We [patients] pay a small amount of money and get the medicines we did not get at the facility pharmacy. At times the strips which we use to check our sugar levels are out of stock, so they use the money to buy the strips.* (IDI, Public, Mbarara, male, 63 years).

*The chairperson often puts pressure on the doctors to ensure that we get drugs. She can go to the medical superintendent and demand for the medicines. Sometimes she borrows, if our contribution was not enough. She also gives us advice on how to survive as patients with diabetes.* (FGD, Public, Entebbe, female, Age 51 years).

The association leaders supported these comments regarding their advocacy on behalf of patients:

*Even when the money is little, we still get the medicines and promise to pay later when we get the funds. We go to the suppliers to make our request. We ensure to get enough drugs and promise to pay the debt say like in three months or three weeks depending on our income.* (KII, Association Chairperson, Mbarara).

*I sought for measures to see to it that we can lobby for medicines because they are not enough as I told you. So in collaboration with the basawo* [HCW], *we were able to get attention from government and we frequently get supplied with medicines. Even though at times medicines run out of stock, it is not as frequent as it is with other hospitals. We also mobilize money to buy medicines as a group but it is not easy to get enough money.* (KII, Association Chairperson, Public, Entebbe).

## Discussion

This qualitative study explored barriers and facilitators of access to medicines to treat diabetes and hypertension at public and private sector health facilities in Uganda. Barriers and facilitators are summarized in four thematic areas: Stocking of medicines and supplies, Financial factors, Individual behaviour and attitudes, and Service delivery at health facilities.

Management of NCDs requires long term, often lifetime, adherence to medicines. Such adherence, in part, relies on consistent stock of these medicines at pharmacies. In our study, although the presence of free medicines was a facilitator of access and a motivating factor for participants to seek treatment at public facilities, their frequent stockouts presented a barrier to access. This finding is in concordance with other study findings in Uganda [4, 5, 14], which report low availability of NCD medicines at public facilities. Our quantitative work performed in parallel to this study also established that a majority of medicine doses prescribed for diabetes and cardiovascular disease at Ugandan public sector facilities are not dispensed [15]. This represents a major challenge to adherence, as public facilities are the primary source of free medicines for patients. Reasons for stockouts of medicines at public facilities are multi-factorial and can be attributed to inaccurate forecasting of demand by facility staff, inefficient procurement or distribution mechanisms in the supply chain and inadequate budget allocation from Ministry of Health to facilities [16]. Additionally, medicines in the public sector are distributed to facilities in four-to-eight-week cycles, resulting in wide fluctuations in availability [17]. There are ongoing efforts to improve forecasting of NCD medicines in order to improve facility-level availability. One such effort, in direct collaboration with Ministry of Health Uganda is the PATH Coalition for Access to NCD Medicines and Products [18]. This coalition of government agencies, private-sector entities, nongovernment organizations philanthropic foundations, and academic institutions is focussed on increasing access to NCD medicines and health products through developing improved forecasting tools, strengthening supply chains, and strengthening health facility capacity. In our study, due to poor availability, medicines at some public facilities were rationed to ensure their availability for certain vulnerable

sub-populations which facilitated access to these populations. While not an effective strategy for improving the health of the population, this may serve to improve access to those at the extremes of age or impoverishment.

When medicines are unavailable, patients are instructed to purchase them at private pharmacies, though not all patients could afford to purchase these medicines. Medicines essential in the management of diabetes and hypertension are generally unaffordable for many Ugandans [4, 14]. Our parallel quantitative study identified that only 36% of patients who do not receive their medicine doses in public facilities actually purchased these medicines [15]. This challenges the underlying assumption held by many healthcare workers that patients are topping off their under dispensed medicines through purchasing in the private sector. Aside from the lack of funds to afford medicines, a lack of resources to pay for transport was another barrier to reaching care and obtaining appropriate medicines. Previous literature demonstrates that a lack of money for healthcare and transport is a barrier to accessing medicines [19, 20]. This study reveals that health insurance, though still limited in settings such as Uganda, serves as both a barrier and facilitator to medicine access. More research is needed to understand the implications of insurance coverage limits on the continuity of care and medicine adherence in such settings.

Our findings demonstrated multiple barriers to medicine access related to individual behaviour and attitudes. Despite the provision of free services at public facilities, patients' negative perception of free medicines contributed to reduced access and utilization. Among the concerns over free medicines was the quality of particular generic insulin brands. The variable quality of the global generic medicine supply is a widely recognized issue. Though we recognize the inherent challenges presented by monitoring the quality of generics, governments and private sector entities should work toward procuring medicines from companies that have demonstrated good manufacturing practice, high quality, and bioequivalence [21]. Low quality drives patients' negative perceptions of medicines which, as we demonstrate, drives mistrust and non-adherence. Providers should work to address patients' negative beliefs about medicines in order to improve adherence [22]. Future work should aim to more fully understand the ways in which patients in LMIC navigate these complexities and make choices about seeking NCD care.

We found that patients seek guidance from several providers and their peers, resulting in mixed messages about medicine safety and effectiveness. The seeking of competing opinions may be attributed to patients' lack of information and brief patient–provider interactions where patients neither ask nor are encouraged to ask questions [23]. Given the silent nature of NCDs, educating patients about their health conditions and their prescribed treatments is imperative. This may require a team-based approach and multiple educational modalities for public sector facilities which have high patient volume. A recent assessment demonstrates that less than 10% of Ugandan public GHs had an NCD counsellor or nutritionist on staff or even offered NCD informational materials. However more than half of these facilities offered individual or group NCD education and about one-third offered peer group NCD support [11]. More research is needed to determine effective multi-modal strategies to provide continuing NCD education in resource-constrained environments.

Some of our participants used herbal treatments and even vegetables as alternatives or supplements to prescribed medicines. The use of herbal treatments was often viewed as an adaptation due to difficulty accessing hospitals and medicines, especially among patients attending public facilities. However, as others have also demonstrated, some patients believe so strongly in the intrinsic effectiveness of herbal treatments, that they preference these treatments over medicines regardless of medicine access [20, 24, 25]. Use of herbal treatments in this region is common and is influenced by peers, family members as well as traditional healers [24]. Future

work should seek to generate empiric data about these local herbs and vegetables and their health effects so providers can be better informed about how to recommend for or against their use and patients can be given accurate information.

Our final theme centered around the provision of health services which largely included barriers in public facilities and a facilitator in private facilities. Since patients pay for services at private facilities, these facilities are seen to be more efficient and accountable, and to potentially provide higher quality care than those in the public sector [26]. Service delivery at public facilities is characterised by restricted operating hours for NCD care on specific days, high patient volumes, and HCW burnout, which some patients interpreted as rudeness and lack of interest in their care. All of these factors present barriers to NCD medicine access. With the objective to avail care to a wider population, public facilities may fail to equitably serve their patients [27] unless the care structure and human resource allocation are improved to lessen health worker overload. The presence of high patient volumes at public facilities can, however, be utilized to improve patient education and engagement by clustering staff educational efforts and through engagement with peers who have the same chronic conditions. Despite the barriers presented, an intriguing finding of this research was the existence of diabetes patient associations at some public facilities and their role in mitigating medicine shortages and advocating for higher quality patient care. Patient driven-groups have been previously reported to support and extend existing health care delivery models [28]. Facility-based groups are among the options offered to patients with stable HIV through the Differentiated Service Delivery platform in Uganda and other LMIC [29, 30]. We are currently developing models of facility-based groups for hypertension within an integrated hypertension/HIV care platform in Uganda's largest HIV clinic. What differentiates these diabetes associations is that they are fully patient-driven. Great opportunity exists to further develop and comprehensively study the role of patient groups in improving the quality of NCD care.

This work should be regarded within the context of some limitations. The public facilities considered were only hospitals, higher level facilities which are typically better resourced than lower level health facilities. Therefore, patient experiences might be different when comparing those who seek care at public hospitals and those who attend lower level health facilities. Our PFP sample was limited to facilities in the IMC network. Since service provision among PFP facilities differ, the barriers and facilitators may not fully reflect what patients experience at other PFP facilities. However, we aimed for achieving validity in this study by triangulating data collection methods (FGD, IDI, and KII) and by obtaining data from a maximum variation sample that included patients, group leaders, and HCW. Similarly, our analytic strategy included a multidisciplinary coding team, forward and backward movement across the data, cross-linkage between codes and categories, and links between the original data and findings until coherent themes emerged, allowed flexibility, divergent ideas and reduced biases.

## Conclusion and recommendations

As LMICs such as Uganda restructure their health systems to address the growing NCD epidemic, both patient—and health system-level barriers to medicine access should be addressed and facilitators should be leveraged. Our findings demonstrate the interaction of barriers and facilitators to accessing medicines at both public and private facilities. These findings can provide a basis for future research and translation into policy and practice. Patient-driven associations could potentially catalyse improved access to medicines in the public sector and could expand in scope beyond diabetes. The inconsistency of availability in the public sector and affordability of medicines in the private sector should be addressed. It is unlikely that international donor funding, which has ensured access to antiretroviral medicines for HIV medicines

through vertical funding mechanisms, will manifest for NCD medicines. Similarly, universal health coverage schemes are also unlikely to suffice to fully cover the cost of NCD medicines for the population. However, potential solutions, such as public-private partnerships, local generic pharmaceutical production, and reduced pricing pharmaceutical programmes could address the identified barriers. Finally, innovative platforms for patient-centered NCD service delivery as well as multi-modal patient education strategies could serve to break down the multiple barriers to NCD medicine access.

## Supporting information

**S1 File. Interview guides.**
(DOCX)

**S2 File. COREQ checklist.**
(PDF)

## Acknowledgments

The authors would like to acknowledge the following individuals for their support of this research: the District Health Officers and health facility administrators who allowed us to conduct this research within the public sector facilities; Rose Nanyonga-Clarke, PhD, Vice Chancellor, Clarke International University for establishing linkages with the International Medical Centre network of private facilities; and our research Assistant, Annette Nabunya, for her dedication to ensuring high quality in this work.

## Author Contributions

**Conceptualization:** Andrew K. Tusubira, Isaac Ssinabulya, Christine K. Nalwadda, Jeremy I. Schwartz.

**Data curation:** Andrew K. Tusubira, Brenda D. Nakirya, Ritah Nalwoga, Christine K. Nalwadda.

**Formal analysis:** Andrew K. Tusubira, Ann R. Akiteng, Brenda D. Nakirya, Ritah Nalwoga.

**Funding acquisition:** Jeremy I. Schwartz.

**Investigation:** Andrew K. Tusubira, Ritah Nalwoga.

**Methodology:** Andrew K. Tusubira, Brenda D. Nakirya, Ritah Nalwoga, Isaac Ssinabulya, Christine K. Nalwadda, Jeremy I. Schwartz.

**Project administration:** Andrew K. Tusubira, Ann R. Akiteng.

**Resources:** Ann R. Akiteng, Isaac Ssinabulya, Christine K. Nalwadda, Jeremy I. Schwartz.

**Software:** Andrew K. Tusubira, Ann R. Akiteng, Brenda D. Nakirya, Ritah Nalwoga.

**Supervision:** Isaac Ssinabulya, Christine K. Nalwadda, Jeremy I. Schwartz.

**Validation:** Christine K. Nalwadda, Jeremy I. Schwartz.

**Visualization:** Andrew K. Tusubira.

**Writing – original draft:** Andrew K. Tusubira, Christine K. Nalwadda, Jeremy I. Schwartz.

**Writing – review & editing:** Andrew K. Tusubira, Ann R. Akiteng, Brenda D. Nakirya, Ritah Nalwoga, Isaac Ssinabulya, Christine K. Nalwadda, Jeremy I. Schwartz.

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
