## [Decision Letter · Decision Letter 0]

24 Feb 2020

PONE-D-20-02052

Accessing Medicines for Non-Communicable Diseases: Patients and Health Care Workers’ Experiences at Public and Private Health Facilities in Uganda

PLOS ONE

Dear Mr Tusubira,

Thank you for submitting your manuscript to PLOS ONE. After careful consideration, we feel that it has merit but does not fully meet PLOS ONE’s publication criteria as it currently stands. Therefore, we invite you to submit a revised version of the manuscript that addresses the points raised during the review process.

We would appreciate receiving your revised manuscript by Apr 09 2020 11:59PM. To enhance the reproducibility of your results, we recommend that if applicable you deposit your laboratory protocols in protocols.io, where a protocol can be assigned its own identifier (DOI) such that it can be cited independently in the future. For instructions see: http://journals.plos.org/plosone/s/submission-guidelines#loc-laboratory-protocols

We look forward to receiving your revised manuscript.

Kind regards,

Khin Thet Wai, MBBS, MPH, MA (Population & Family Planning Resear

Academic Editor

PLOS ONE

Additional Editor Comments (if provided):

This is an important study addressing the issue related with access to essential medicines for NCDs covering both public and private sectors. To strengthen scientific integrity and applicability of research evidence for Universal Health Coverage, authors have to revise extensively. Moreover, English language correction is deemed necessary.

Journal Requirements:

Reviewers' comments:

Reviewer's Responses to Questions

**Comments to the Author**

1. Is the manuscript technically sound, and do the data support the conclusions?

Reviewer #1: Yes

Reviewer #2: Yes

2. Has the statistical analysis been performed appropriately and rigorously? 

Reviewer #1: Yes

Reviewer #2: Yes

3. Have the authors made all data underlying the findings in their manuscript fully available?

Reviewer #1: Yes

Reviewer #2: Yes

4. Is the manuscript presented in an intelligible fashion and written in standard English?

Reviewer #1: No

Reviewer #2: No

5. Review Comments to the Author

Reviewer #1: Thank you for giving me the opportunity to review this manuscript.

The manuscript is well written and addresses an important public health problem. However, the manuscript requires major revision in terms of methods and results section.

General comments: Entire manuscript requires proofreading for typographical and grammatical errors.

Introduction: Background and rationale are well written. However, further points on the necessity and application of current study findings is required.

Methods: Study setting: Details on distribution of NCD patients receiving care at public sector and private sector can be provided. Public and private sector distribution can be provided.

Study design: Can be mentioned as qualitative descriptive instead of cross-sectional qualitative

Sampling strategy: total number of facilities in this region can be provided to see how much the sample is representative. Case load of this hospitals can also be provided. Maximum variation sampling can be mentioned here as it comes up only in overcoming the limitation part

Table 1: Why there is over-representation of public health facilities and only 2 FGD done in private health facilities.

Data collection and management: Details on why these many number of FGD, IDI and KII was done can be mentioned. Was it done till data saturation or pre-fixed numbers?

Table 3: Framework can be split into two: As patient perspective and health provider prespective as it will help in exploring both their perspectives. Barriers and facilitators is also not differentiated and mixed up in the same framework. Hence, under each theme, barriers and facilitators can be stated in bracket.

Discussion: Discussion is very well written by summarizing the main study findings and comparing and contrasting with previous studies with relevant recommendation for public health action and future studies. Limitations in the study is also provided.

However, a small suggestion is that the author can consider removing the second paragraph in discussion can be removed as it anyways gets repeated throughout the discussion part.

Reviewer #2: This is an important study in the under-examined field of medication access in Uganda. As chronic diseases take over the disease burden in this country and the surrounding region - concurrent with a rise in private-sector care - examination of barriers and facilitators to obtaining medications for non-communicable disease in private and public facilities becomes ever more urgent. I feel the study is robust and its conclusions rigorous and easy to follow. My only comments - for a suggested minor revision - relate to clarity of writing and some remaining questions regarding the methods and implications of the work. Details by manuscript line are as below:

Abstract: would remove "interestingly" from the results section (line 55) and expound a bit more on actionable next steps in the conclusion section (lines 59-60) - for instance why these three proposed solutions are most likely to succeed across both public and private sectors. Would add commas to line 51.

Methods: would change to "works on a referral basis" in line 87. Also kindly clarify the "apparently influence of diabetes patient groups" in lines 109 - 111: who are these groups; how do they influence medication access in Uganda; and what did you feel that their chairpersons would add to the study if interviewed? Lines 129-130 allude to this issue but the specifics remain unclear. How did you select the hospitals and clinics (line 114) apart from geographic diversity? Also, how did you decide (lines 126-127) which patients were assigned to an FGD or IDI? Did patients get to select one of these options, or did you do so? Lastly, please clarify the Framework Method further in lines 156-159 - how does it differ from other analytic strategies in which themes should be analyzed concurrent with contextual review?

Results: generally clear and straightforward, in my view. I would change some passive-voice sentences (lines 246, 339) to active for readability.

Discussion: the first few summary lines (400-404) are somewhat redundant and could be shortened or omitted. Similarly, the paragraph from 405-412 - which largely states results rather than analyzing them - could be deleted or moved to the results section. More generally, I'd be excited to see the discussion focus more on potential solutions than the underlying cause of the problems with access. What, for instance, is the PATH Coalition doing now, and what might it do better going forward, in promoting NCD medication access in Uganda? The final paragraph from lines 517 to 532 discusses such steps, but would benefit from more detail if possible - for instance around the nature of innovative patient-centered delivery platforms as mentioned in lines 496-499. Lastly, please note typos in lines 430 and 439.

Thank you for the opportunity to review this important work. I'd be happy to review a revised draft if helpful.

6. PLOS authors have the option to publish the peer review history of their article (what does this mean?). If published, this will include your full peer review and any attached files.

Reviewer #1: No

Reviewer #2: No

---

## [Author Response · Author response to Decision Letter 0]

6 May 2020

Dear Editor,

Our response letter has been uploaded along with the other attachments. 

Thank you, 

Andrew.

---

## [Decision Letter · Decision Letter 1]

22 Jun 2020

Accessing Medicines for Non-Communicable Diseases: Patients and Health Care Workers’ Experiences at Public and Private Health Facilities in Uganda

PONE-D-20-02052R1

Dear Dr. Tusubira,

We’re pleased to inform you that your manuscript has been judged scientifically suitable for publication and will be formally accepted for publication once it meets all outstanding technical requirements.

Kind regards,

Khin Thet Wai, MBBS, MPH, MA (Population & Family Planning Resear

Academic Editor

PLOS ONE

Additional Editor Comments (optional):

Reviewers' comments:

Reviewer's Responses to Questions

**Comments to the Author**

1. If the authors have adequately addressed your comments raised in a previous round of review and you feel that this manuscript is now acceptable for publication, you may indicate that here to bypass the “Comments to the Author” section, enter your conflict of interest statement in the “Confidential to Editor” section, and submit your "Accept" recommendation.

Reviewer #2: All comments have been addressed

2. Is the manuscript technically sound, and do the data support the conclusions?

Reviewer #2: Yes

3. Has the statistical analysis been performed appropriately and rigorously? 

Reviewer #2: I Don't Know

4. Have the authors made all data underlying the findings in their manuscript fully available?

Reviewer #2: Yes

5. Is the manuscript presented in an intelligible fashion and written in standard English?

Reviewer #2: Yes

6. Review Comments to the Author

Reviewer #2: I appreciate the authors' dutiful response to my critique - which concerned chiefly readability rather than content. However, the authors have also carefully addressed important questions from reviewer 1 regarding the pertinent background and context, as well as the distribution of facilities studied. I have no further comments to add and recommend the paper is accepted and published as written.

7. PLOS authors have the option to publish the peer review history of their article (what does this mean?). If published, this will include your full peer review and any attached files.

Reviewer #2: No

---

## [Editor Report · Acceptance letter]

25 Jun 2020

PONE-D-20-02052R1 

Accessing Medicines for Non-Communicable Diseases: Patients and Health Care Workers’ Experiences at Public and Private Health Facilities in Uganda 

Dear Dr. Tusubira:

I'm pleased to inform you that your manuscript has been deemed suitable for publication in PLOS ONE. Congratulations! Your manuscript is now with our production department. 

Kind regards, 

on behalf of

Dr. Khin Thet Wai 

Academic Editor

PLOS ONE